# Extracellular ATP as an Inter-Kingdom Signaling Molecule: Release Mechanisms by Bacteria and Its Implication on the Host

**DOI:** 10.3390/ijms21155590

**Published:** 2020-08-04

**Authors:** Daniel Spari, Guido Beldi

**Affiliations:** 1Department for Visceral Surgery and Medicine, Bern University Hospital, University of Bern, CH-3010 Bern, Switzerland; daniel.spari@dbmr.unibe.ch; 2Department for BioMedical Research (DBMR), Visceral and Transplantation Surgery, Bern University Hospital, University of Bern, CH-3008 Bern, Switzerland

**Keywords:** bacterial ATP, bacteria-derived ATP, bacteria, extracellular ATP, ATP secretion, purinergic signaling, inflammation, sepsis

## Abstract

The purine adenosine 5′-triphosphate (ATP) is not only a universal intracellular energy carrier but plays also an important role as extracellular signaling molecule. Purinergic signaling is involved in many physiological and pathological processes like coagulation, inflammation, or sepsis in mammals. ATP is well-known as a messenger for intercellular communications in multicellular organisms, but phylogenetically much older unicellular organisms like yeast or bacteria use ATP as an extracellular signaling molecule as well. However, the mechanisms of ATP secretion by bacteria and its extracellular implications still have to be elucidated. This review will provide an overview of the current knowledge about bacterial extracellular ATP (eATP) under homeostatic conditions and during growth. Possible secretion mechanisms of ATP by bacteria will be discussed and implications of bacterial ATP are shown, with a focus on bacteria–host interactions.

## 1. Principles of Purinergic Signaling

While the importance of the purine adenosine 5′-triphosphate (ATP) as a universal intracellular energy carrier was discovered in the middle of the 20th century, the landmark discoveries about the importance of ATP as an extracellular signaling molecule took another 20 years [1]. Intracellularly, ATP enables energy-dependent processes releasing high-energy electrons by the hydrolyzation of its three phosphoanhydride bonds. This intracellular ATP can be released into the extracellular space by specific or unspecific mechanisms and acts as a danger associated molecular pattern (DAMP) [2], affecting platelet activation and coagulation [3,4], inflammation [5,6,7], and a wide range of neural processes [8].

Extracellular ATP (eATP) binds to specific P2 receptors, which are grouped into ionotropic P2X (P2X1, P2X2, P2X3, P2X4, P2X5, P2X6, P2X7) and metabotropic (G-protein-coupled) P2Y (P2Y_1_, P2Y_2_, P2Y_4_, P2Y_6_, P2Y_11_, P2Y_12_, P2Y_13_, P2Y_14_) receptors [9,10]. After having bound eATP, P2X receptors undergo conformational change and facilitate the influx as well as efflux of mono- and divalent cations along their concentration gradient (mainly Na^+^, Ca^2+^ in and K^+^ out) [11]. P2Y receptors activate either stimulatory or inhibitory G proteins, which mediate downstream signaling regulating 3′,5′-cyclic adenosine monophosphate (cAMP) and phospholipase C β (PLCβ) levels [12]. While P2X receptors are exclusively modulated by ATP, ligands for P2Y receptors also include adenosine 5′-diphosphate (ADP), uridine 5′-triphosphate (UTP), uridine 5′-biphosphate (UDP), and UDP-glucose [10,12]. Extracellular ATP is rapidly degraded by ectonucleotidases that convert ATP to ADP and adenosine 5′-monophosphate (AMP) as well as CD73/ecto-5′-nucleotidase that converts AMP to adenosine [13]. Adenosine is able to bind four distinct receptors: A_1_, A_2A_, A_2B_, or A_3_ [14]. All four are metabotropic (G-protein-coupled) receptors transducing extracellular signals mainly via regulation of cAMP [15]. In contrast to ATP receptors, adenosine receptors play a pivotal role in attenuating inflammation [16,17,18]. This dynamic regulation allows purinergic signaling to be both pro-inflammatory (mainly but not exclusively via P2 receptors) and anti-inflammatory (mainly but not exclusively via P1 receptors) [19,20,21]. Furthermore, the ligand binding affinities for different P2X receptors are within the nanomolar to the high micromolar range, making purinergic signaling a highly versatile communication system [22].

## 2. Secretion and Uptake of ATP by Microorganisms

The importance of the purinergic communication system in multicellular organisms like animals or plants has been established over the last decades. However, not only multicellular but also phylogenetically much older unicellular organisms make use of purinergic signaling or are at least able to secrete ATP [23].

These includes fungi such as *Candida albicans* (*C. albicans*) that secretes ATP putatively via conductive channels in a histatin 5-dependent manner [24]. Such eATP subsequently leads to cell death in neighboring *C. albicans* cells by a hitherto unknown mechanism [24]. The unicellular yeast *Saccharomyces cerevisiae* (*S. cerevisiae*) secretes ATP in the nanomolar range in a *MCD4*-dependent manner [25]. Incubation of *S. cerevisiae* with 2% glucose solution dramatically increased eATP concentration in a cAMP-dependent and regulated manner [26]. Furthermore, eATP was elevated during growth of *S. cerevisiae* [26]. The conclusion of these observations was that ATP, its metabolites ADP, AMP, and adenosine may act as intercellular signaling molecules for *S. cerevisiae* physiology [26]. Extracellular ATP as an intercellular messenger in *S. cerevisiae* has also been shown in synchronizing sporulation among cells, yet the mechanism has still to be elucidated [27].

Not only such unicellular eukaryotes though, but also prokaryotes are able to secrete ATP and there is now an increasing number of reports focusing on bacterial ATP, the invading bacteria as relevant source of eATP in the host and its importance in bacteria–host interaction [28].

### 2.1. Bacterial eATP under Homeostatic Conditions

In bacteria, first results were obtained in 2005 by Ivanova et al. [29]. A screen of the supernatant of 86 heterotrophic environmental (marine) bacteria cultured in Marine Broth 2216 for eATP showed that Gram-negative α-proteobacteria and γ-proteobacteria secreted ATP in the micro- to millimomolar range in vitro. The measured concentrations were between 0.2 and 7 mM [29,30,31]. Overall, marine Gram-positive bacteria (*Planococcus* spp., *Kocuria* spp., *Brevibacterium* spp., *Bacillus* sp., *Microbacterium* sp.) secreted less ATP than marine Gram-negative α-proteobacteria (*Ruegeria* spp., *Erythrobacter* sp., *Sulfitobacter* spp., *Staley* sp.) or γ-proteobacteria (*Marinobacter* spp., *Marinobacterium* sp., *Marinobulbifer* sp., *Cobetia* spp., *Alteromonas* spp., *Pseudoalteromonas* spp., *Shewanella* spp.) [29]. The concentrations in the supernatant even exceeded values that were measured inside cells [32,33,34]. In vivo, the ability of commensal bacteria to secret ATP was assessed from isolates from mouse feces and from human feces, urine, and skin. In mouse feces, *Enterococcus gallinarum* (*E. gallinarum*), *Enterococcus faecalis* (*E. faecalis*) and *Escherichia coli* (*E. coli*) and in human specimens, *E. gallinarum*, *E. faecalis*, *E. coli*, and in addition *Enterococcus faecium* (*E. faecium*) as well as *Staphylococcus aureus* (*S. aureus*) were cultured in RPMI 1640 medium and eATP levels in the supernatant were assessed after 16 h [35]. Thereby, eATP at levels between 1 and 3 µM were detected only in the supernatant of Gram-positive *E. gallinarum* culture, whereas in the supernatant of the other bacterial cultures, eATP was absent [35].

### 2.2. Bacterial eATP During Growth

In the follow-up study by Hironaka et al., seven more ATP secreting *Enterococcus* species were identified (*E. cecorum*, *E. faecium*, *E. gilvus, E. mundtii*, *E. saccharolyticus*, *E. sulfureus*, and *E. thailandicus*) under the same culture conditions as described above [36]. As in the previous study, eATP concentration was in the low µmolar range (around 0.1–2.2 µM/10^8^ cells), and highest in *E. mundtii*. Importantly, these studies revealed that ATP secretion is growth phase-dependent, peaking during exponential phase. These observations allowed to detect eATP also in the supernatant of *E. faecalis, E. faecium*, *E. coli*, and *S. aureus* during exponential growth phase [36]. According to the authors, checking membrane integrity using a Sytox9/propidium iodide based staining excluded bacteriolysis as a source of ATP secretion [36].

Growth-dependent ATP secretion during exponential phase in the supernatant of a variety of different Gram-positive and Gram-negative bacteria was also observed by Mempin et al., however, in the low nanomolar range [37]. A possible explanation for the discrepancy in the amount of eATP between these two groups is that Hironaka et al. used RPMI 1640 medium for culturing bacteria, which contains a different amount of ATP secretion triggering glucose than standard Luria–Bertani broth (LB), which was used by Mempin et al. [36]. Another possibility is that the used RPMI 1640 medium was enriched with serum, which accounted for the higher amount of ATP. However, this is rather unlikely since the authors did not mention any supplementation with serum in their paper. The question remains, why Hironaka et al. did not observe ATP secretion with the tested media other than RPMI 1640 medium (e.g., brain heart infusion (BHI), LB, tryptic soy agar (TSA)), since the group used the same reagents to measure eATP like Mempin et al. This could be explained by different methods they used for the generation of their respective standard curves allowing Mempin et al. more sensitive measurements and therefore to detect lower eATP concentrations than Hironaka et al.

In addition to glucose, cytochrome *bo* oxidase plays an important role for eATP concentration in *E. coli* and *Salmonella enterica* (*S. enterica*) [37]. Cytochrome *bo* oxidase subunits mutated *E. coli* (*ΔcyoA*, *ΔcyoC*, *ΔcyoD*) and *S. enterica* (*ΔcyoA*, *ΔcyoB*, *ΔcyoCD*) showed significantly less eATP compared to the wild type when cultured aerobically [37]. Cytochome *bo* oxidase is one of three different ubiquinol oxidases, enzymes that pump protons across the inner membrane of the bacteria generating a proton motif force and that catalyzes the last step of the respiratory chain, which is the reduction of the electron acceptor. Furthermore, cytochrome *bo* oxidase predominates at high oxygen concentrations [38,39]. Together with the results from Hironaka et al. this could indicate that in *E. coli* and *S. enterica*, glycolysis and subsequent anaerobic or aerobic respiration are more important than fermentation for ATP secretion, which makes sense, since it is much more effective for generating ATP and bacteria can use a variety of redox pairs for respiration [36,37,40].

### 2.3. Uptake of eATP by Bacteria and Intracellular Fate

Several reports show that bacteria are also able to take up ATP according to their needs. In an experiment using *E. coli W* derived strains that have mutations in purine metabolism, it was shown that these bacteria were able to take up a variety of different purine nucleobases, nucleosides, and nucleotides (adenine, guanine, adenosine, guanosine, AMP, ADP, and ATP) possibly via general porins, which restored growth deficiency [41]. Bacteria mediated depletion of ATP was also shown in an experiment, in which 10 µM ATP was added to the bacterial culture and the remaining concentration was measured after several time intervals [37]. An almost complete depletion of the 10 µM was achieved by *E. coli K12* as well as *S. enterica (SE2472*) within 120 min [37]. Interestingly, they concluded that ATP was mainly hydrolyzed on the bacterial surface, given that the remaining levels of radioactivity were detected in the supernatant and not in the bacteria. Conversely, using radioactively labelled [γ-32P]ATP, Alvarez et al. showed that in *E. coli DH5α* ATP is rapidly uptaken in the periplasmic space and hydrolyzed in a linear [eATP]-dependent manner and that eATP concentration in steady state in the periplasmic space is 24 ± 3 µM/10^10^ bacteria [42]. The periplasmic space as an important compartment involved in purine hydrolyzation is supported by a variety of scientific reports [41,42,43] and an array of acid and alkaline phosphatases (*aphA*, *appA*, *phoA*) as well as 5′-nucleotidases (e.g., *napD*, *ushA*, *surE*, *yfbR*, *yjjG*) are present in *E. coli*. Under physiological conditions especially *phoA* encoded alkaline phosphatase and *ushA* encoded 5′-nucleotidase are responsible for ATP hydrolyzation to adenosine [42,43,44,45,46]. Adenosine can then be further metabolized to adenine or hypoxanthine or can be directly transported back to the cytoplasm via the specific nucleoside transport systems *nupC* and *nupG* and recycled to ATP [41,47].

Interestingly, it is still unclear how ATP is transported from the cytosol to the periplasmic space since *E. coli* do not have ATP/ADP translocases in the inner membrane and this membrane is impermeable for the highly (four-fold) negatively charged ATP [43,48]. Furthermore, it still has to be determined how bacteria secrete ATP and possible mechanisms will be discussed in the next section.

## 3. Comparison of ATP Secretion Mechanisms between Prokaryotic and Eukaryotic Cells

### 3.1. Release Mechanisms of Eukaryotic Cells

In eukaryotic cells, there are mainly three distinct mechanisms of ATP release: (1) channel-mediated release, (2) vesicle-mediated release, and (3) non-specific release [49].

(1) Channel-mediated ATP release occurs mainly via pannexin-1 channels and connexin-43 hemichannels [12,49]. Pannexin-1-dependent ATP release occurs in apoptotic cells, which thereby establish a pro-inflammatory environment and attract leucocytes by releasing ATP in a controlled manner [50]. Pannexin-1 is also involved in ATP release by dendritic cells and subsequent autocrine signaling via P2X7 that creates a positive feedback loop, promoting fast migration at site of inflammation [51]. Additionally, T-cells are activated via pannexin-1-mediated ATP release and autocrine P2X1 and P2X4 signaling at the immune synapse, which represents a general T-cell activation mechanism [52]. Furthermore, macrophages can be activated by pathogen associated molecular patterns (PAMPs) to release ATP via pannexin-1, autocrine signaling via P2X7, and subsequent inflammasome activation [53]. Connexin-dependent ATP release occurs in neutrophils and macrophages upon stimulation with PAMPs [30,31,49]. This ATP release has two different effects: (1) it supports the inflammatory milieu [6] and (2) it stops neutrophil chemotaxis at site of inflammation in an autocrine manner [54]. Macrophages also release ATP via Connexin-43 in an autocrine manner, resulting in self-activation and improved survival during sepsis in mice [30].

(2) Vesicle-dependent release requires storage of ATP in vesicles, which is mediated by a V-ATPase dependent proton gradient [49]. V-ATPase is localized in the vesicle membrane and actively pumps H^+^ into the vesicle lumen. Negatively charged ATP is subsequently transported into the lumen via vesicular nucleotide transporter (VNUT) along the electrochemical gradient [55,56]. Release of vesicular ATP is then mediated by Ca^2+^-dependent zipper-like fusion of *v-* and *t-*SNARE complexes localized in the vesicle and the plasma membrane, respectively [57]. Vesicular ATP release has been shown to play an important role in mediating chronic inflammatory pain [58]. Different cells (e.g., microglia or peritoneal macrophages) use this release mechanism upon stimulation with PAMPs [58,59,60].

(3) Non-specific release of ATP typically depends on necrosis of cells. The intracellular ATP concentration is within the millimolar range, which is more than 10^5^-fold higher than extracellular ATP concentration [6,61]; therefore, disruption of membrane integrity leads to an instant release of high amounts of ATP. This establishes a strong pro-inflammatory environment and a strong chemotactic gradient [5,12].

### 3.2. Release Mechanisms of Prokaryotic Cells

In bacteria, the detailed mechanisms of ATP secretion are less known as in eukaryotic cells. Bacteria do not possess pannexin or connexin hemichannels nor are any homologues of VNUT known. However, some authors have described and shown different mechanisms how ATP can be secreted: (1) by porins, (2) by mechanosensitive (MS) channels, or (3) by outer membrane vesicles (OMVs). Another possibility is that ATP is secreted by (4) lytic mechanisms especially during growth. Furthermore, it is conceivable that bacterial efflux pumps or secretion systems (5) are responsible for ATP secretion although such specific pumps for ATP have not been shown so far [62,63,64,65,66,67] (Figure 1).

(1) Porins are water-filled β-barrel pores in the membrane of Gram-positive or the outer membrane of Gram-negative bacteria that are permeable to water-soluble molecules. In *E. coli K12*, there are specific porins, e.g., *tsx*, which transports nucleosides and deoxynuclosides [68,69] or *lamB*, which transports maltose and maltodextrins [70]. However, there are also three unspecific porins called *phoE*, *ompC*, and *ompF* that are permeable for solutes smaller than 600–700 Dalton. *PhoE* is especially induced by phosphate starvation and has an anion selectivity, whereas *ompC* and *ompF* are rather cation-selective [71]. So far, no specific porin for nucleotides has been characterized and it is assumed that ATP is unspecifically secreted by these porins [72,73]. It is currently not known if there is a mechanism that allows *E. coli* to control the transport of ATP through the pore or if it is just passively governed by tonicity.

(2) It has been shown that if bacteria are exposed to hypotonic osmotic shock, they secrete ATP and other metabolites and ions via MS channels to prevent bursting [74,75]. However, if this mechanism is involved or responsible for the measured eATP concentrations during growth remains to be elucidated.

(3) OMVs are produced and released especially by Gram-negative bacteria and play a role in promoting pathogenesis, bacterial survival, and interaction [76]. They pinch off the outer membrane of Gram-negative bacteria and therefore contain periplasmic solutes. It was shown that OMVs contain also ATP, which they secret in a concentration of 13 ± 5 nM/µg protein [42]. Furthermore, OMVs possess intrinsic linear [eATP]-dependent ATPase activity [42]. That OMVs contain ATP and periplasmic phosphatases was also shown by Proietti et al., however to what extent OMVs contribute to eATP concentration remains unclear [77].

(4) Bacterial cell division is a highly complex mechanism that requires cell division proteins like *Fts-* or *Min.-*proteins, murein hydrolases, and other proteolytic enzymes working together for structured local breakdown of bacterial cell wall [78,79]. Furthermore, cellular integrity is highly dependent on environmental factors like temperature, pH, osmolarity, and concentration of divalent cations [80,81]. It is therefore conceivable that some degree of ATP secretion could happen within the division process because of disturbances in any of these factors. Another reason for eATP could be full lysis of bacteria during growth. It was shown that lysis in growing cultures is a rare phenomenon but since the intracellular ATP concentration is up to 10^5^-fold higher than in the extracellular space, this may be at least partially responsible for eATP concentration [82].

## 4. Extracellular ATP as an Inter-Kingdom Signaling Molecule

Bacterial ATP has been shown to act as an inter-kingdom signaling molecule in different organ systems. Follicular T helper cells (Tfh) in the Peyer’s patches of gut-associated lymphoid tissue are modulated by bacterial ATP [83]. Tfh are key cells in B cell development and subsequent IgA production and need therefore to be tightly regulated, which occurs, at least in part, via P2X7 mediated apoptosis [83]. By gavaging *E. coli* or *S. enterica* either containing an empty plasmid or a plasmid carrying an apyrase that hydrolyzes the secreted ATP and is polarly localized in the periplasmatic space, the role of specifically bacterial ATP could be assessed [84]. Using this approach, Proietti et al. have shown that bacterial ATP leads to P2X7 mediated apoptosis of Tfh, subsequent limitation of IgA secretion from B cells, and therefore a decreased efficiency of oral vaccination [77,83]. Conversely, such limited IgA secretion leads to a diverse microbiota and a favorable metabolic profile (blood sugar levels, body weight) [85].

Atarashi et al. showed that after intraperitoneal and rectal application of 1.25 mg (2.3 µM) ATPγS, the CD70^high^CD11c^low^ intestinal lamina propria cells increase production of IL-6 and IL-23 as well as TGF-β, leading to increased differentiation of naïve helper T cells to IL-17 producing helper T cells (T_H_17) in germ-free ILC mice [86]. The authors stated that this mechanism could be important for physiologic T_H_17 but also for pathologic T_H_17 differentiation, since severe combined immunodeficient mice (SCID-mice) suffered from exacerbated colitis after intraperitoneal application of 1.5 mg (2.9 µM) αβ-ATP [86]. Since germ-free mice had markedly reduced T_H_17 levels and antibiotic treated (vancomycin and metronidazole) specific pathogen free mice showed a significant reduction in T_H_17 and fecal ATP concentration, the group concluded that eATP is derived by commensal bacteria [86].

Abbasian et al. have suggested that ATP secreted by uropathogenic bacteria (e.g., *E. coli*) acts as a virulence factor in urgency urinary incontinence and overactive bladder disease by promoting Ca^2+^ influx into uroepithelial cells affecting subsequent bladder contractility [87].

Since eukaryotic cells are ubiquitously equipped with ecto-nucleoside triphosphate diphosphohydrolases (e.g., CD39), ecto-5′-nucleotidases (e.g., CD73), ectonucleotide pyrophosphatases, as well as alkaline phosphatases and are able to hydrolyze micromolar concentrations of ATP within 20–40 min, ATP signaling is generally considered as fast and short-termed [13,88]. However, the specific kinetics of bacterial ATP under the above-mentioned physiologic and inflammatory conditions still have to be elucidated.

Bacteria are not only able to secrete ATP itself but also to stimulate host cells to release ATP. Alvarez et al. have shown that concentration of apical ATP on a polarized monolayer of Caco-2 cells that was exposed to *E. coli* suspension (2 × 10^7^ bacteria) increased six-fold from basal concentration of 123 ± 5 nM. Furthermore, rat jejunum segments that were flushed with *E. coli* suspensions (5 × 10^8^ bacteria/mL) increased luminal ATP secretion two-fold from basal concentration of 13 ± 2 nM [42]. It has also been shown that the pore-forming toxins α-hemolysin from *E. coli* as well as leukotoxin A from *Aggregatibacter actinomycetemcomitans* lead to ATP secretion of erythrocytes [89]. ATP is most likely released through the toxin pore itself or by the damaged membrane around the toxin and subsequently activates P2X1 and P2X7 receptors, leading to an amplification of the hemolytic process [90].

Furthermore, bacteria seem also to use host-derived eATP for its own purposes. Mempin et al. has shown that addition of 10 or 100 µM ATP to stationary phase *E. coli K12* and *S. enterica* cultures increased survival over 7 days of incubation [37]. The group concluded that this could be of advantage since these bacteria are able to invade host cells, where they could use intracellular ATP as a nutrient or signaling molecule. It has also been shown that eATP promotes biofilm formation in periodontal disease as well as detachment of bacteria from *Fusobacterium nucleatum* biofilm into its planktonic form to build new colonies [91,92]. Extracellular ATP is also involved in maintaining inflammation and bone resorption in periodontal disease via: (1) P2X7 mediated cytokine release by immune cells and subsequent osteoclast activation [93], (2) direct P2X7 mediated fibroblast activation and chemokine production (IL-8, CXCL-1, CXCL-2, and MCP-1) on the mRNA as well as the protein level, and (3) its chemotactic properties as an attractant for leukocytes [94].

Other reports show that bacteria are able to modulate ATP. *Porphyromonas gingivalis*, an intracellular bacterium, expresses a nucleoside-diphosphate kinase to hydrolyze ATP. This bacterium attenuated ATP-P2X7 mediated inflammasome activation and subsequent release of Il-1β as well as release of the DAMP high-mobility group box 1 [95]. *Legionella pneumophila* encodes an ecto-triphosphate diphosphohydrolase, which is similar to human CD39. It has been shown that this enzyme is essential for intracellular multiplication of the bacterium probably also by attenuating ATP-P2X7 receptor mediated inflammasome activation or by attenuating the chemotactic ATP gradient [96]. *S. aureus* and *Bacillus anthracis* use a cell wall anchored 5′-ectonuleotidase to escape phagocytotic clearance during inflammation [97] and *Streptococcus sanguinis* exploits the immunosuppressive ATP to adenosine hydrolyzation of a surface ecto-5′-nucleotidase [98].

## 5. Bacterial ATP as Potential Virulence Factor in Inflammation and Sepsis

ATP has various functions in inflammation and sepsis [5,6,99] and it is therefore conceivable that bacteria exploit the pro-inflammatory properties of ATP for its own purposes or that bacterial ATP interferes with host-derived ATP.

It has been shown that endothelial P2Y_1_ receptor signaling leads to an increased expression of vascular cell adhesion molecule 1 (VCAM-1), intercellular adhesion molecule 1 (ICAM-1), as well as P-selectin, favoring leucocyte rolling and adhesion to endothelial cells [100]. Endothelial cells also actively release ATP after stimulation with bacterial membrane components in a Toll-like receptor 2 dependent way, promoting early immune response [101]. Furthermore, Maître et al. showed that diapedesis of neutrophils is at least partly dependent on ATP–P2X1 interaction [102]. It is therefore conceivable that ATP secreted by bacteria in the blood stream or by bacteria bound to platelets or erythrocytes [103] favors extravasation of leukocytes. Furthermore, it is well known, that some bacteria are able to survive within phagocytes and, therefore, these mechanisms could be exploited by some bacteria for its own dispersal throughout the body [104].

Sepsis is a life-threatening organ dysfunction caused by a dysregulated immune response to invading bacteria, which leads to extensive damage locally but also remote from the primary site of infection [105,106]. Purinergic signaling has been shown to influence septic outcome in various studies [19,21,107,108,109,110,111] and it is therefore conceivable that bacterial ATP is able to activate purinergic receptors and to influence septic outcome as well. It is highly likely that eATP concentration in the surrounding of a bacterium or a bacterial colony is much higher in a halo-like manner than the diluted eATP concentration measured in the supernatant mentioned above. It may therefore be possible that bacterial ATP is able to activate all P2X receptors (even P2X7, which is activated at µmolar concentrations) contributing to the complex system of purinergic signaling.

## 6. Conclusions

ATP is well-known as a major player of the intercellular purinergic signaling system. Purinergic signaling has been extensively investigated especially in multicellular organisms, however there is growing evidence that also phylogenetically much older organisms such as eukaryotic and prokaryotic unicellular organisms use this versatile communication system. Bacteria secrete ATP in a glucose triggered, respiratory chain- and growth-dependent manner. Furthermore, it has been shown that bacteria actively take up and metabolize ATP in the periplasmic space and the cytoplasm. Still a matter of discussion is though, how ATP is transported across the inner membrane from the cytoplasm to the periplasmic space, how ATP is secreted, and if this is an active/specific process. Although eATP has been shown to be an inter-kingdom signaling molecule especially at mucosal sites in the gut or the mouth, the role of bacterial ATP in systemic inflammation and sepsis still has to be resolved and this could lead towards new therapeutic avenues as outlined in the last section of this review.

## Figures and Tables

**Figure 1 ijms-21-05590-f001:**
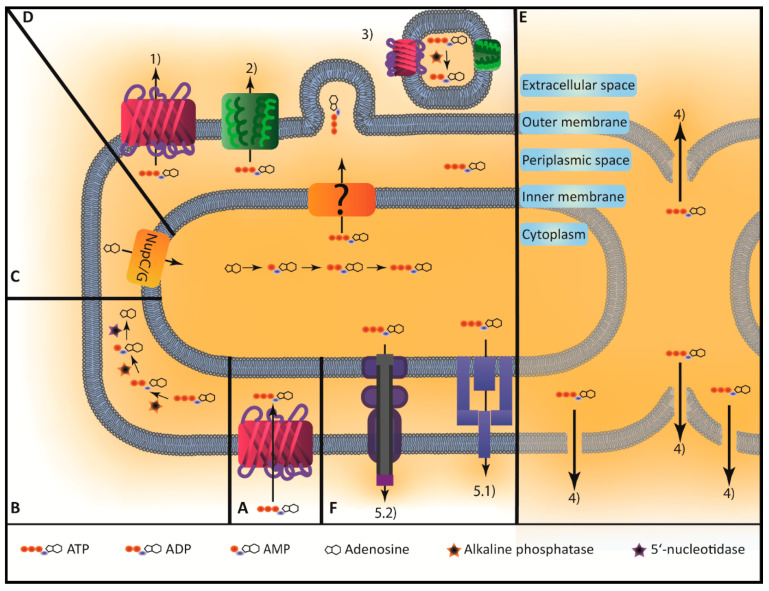
Model of ATP turnover in gram-negative bacteria. (**A**) ATP is taken up by the bacterium, possibly via general porins [41,42]. (**B**) Periplasmic ATP is subsequently hydrolyzed via alkaline phosphatases and 5‘-nucleotidases to ADP, AMP and adenosine [42,43,44,45,46]. (**C**) Adenosine can then be further metabolized or taken up into the cytoplasm via nucleoside permeases like NupC or NupG, where it is recycled to ATP [41,47]. (**D**) ATP is mainly produced in the cytoplasm by the respiratory chain. ATP is then transported into the periplasmic space by a hitherto unknown mechanism (**?**), since as known so far, gram-negative bacteria do not code for an ATP-ADP translocase homologue. Periplasmic ATP can subsequently be secreted in a growth-dependent manner by non-lytic mechanisms such as 1) general porins [42,72,73], 2) mechanosensitive channels [74,75] or 3) outer membrane vesicles [42,76,77]. Most likely ATP surrounds the bacterial cell in a halo-like manner. It has been shown that outer membrane vesicles contain ATP and have intrinsic ATPase activity. (**E**) It is also conceivable that lysis contributes to measured eATP concentrations during growth 4) [78,79,80,81,82]. (**F**) Possibly, active systems such as specific 5.1) efflux pumps or 5.2) secretion systems for ATP exist, but have not been identified so far [62,63,64,65,66,67]. If a specific mechanism is predominating in ATP secretion or if a combination of these mechanims are responsible for bacterial eATP remains to be elucidated.

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
