# Peer review of "Extracellular ATP as an Inter-Kingdom Signaling Molecule: Release Mechanisms by Bacteria and Its Implication on the Host"

_ijms, 2020, doi:10.3390/ijms21155590_

Round 1
Reviewer 1 Report
The Review work by Spari and Beldi focuses on the role of ATP as signal molecules among microorganisms and host cells. The organization of the Manuscript is well done but some new aspects need to be taken into consideration
- As the Authors reported in the Manuscript, production of ATP in microorganisms is highly affected by the culture media. However, the examples reported in the Manuscript concern only in vitro cultured bacteria/fungi. Since in chapter 4, the Authors reported the role of ATP as inter-kingdom signalling molecule, the Authors should report also some evidences about production of ATP by wild-type microorganisms in their natural environment.
- The Authors reported that proximity of bacteria and host cells is pivotal in determining the signal mediated by ATP. Since both microbial and eukaryotic cells are endowed with enzymes involved in ATP degradation, could the Authors make comments about the balance (concentration, time) between degradation and activity of ATP during inflammation?
- Purinergic signal, DAMP and inflammation are usually considered all together. However, ATP is physiologically produced also by non-pathogenic bacteria such as in the gut microbiome. The Authors should refer also to ATP produced by symbiotic bacteria when they consider the ATP as an inter-kingdom signalling molecule.
Minor point
The Authors should better define what they intend as “luminal ATP” referred to Caco-2 cells experiment (lines 241-242)
Lines 43, 66, 218, 219, 253,260: replace “EATP” with “Extracellular ATP”, for clarity
In the legend of Figure 1, I suggest inserting the correspondent citation for references for each described point (A-F)
Sentence lines 181-182 need to be reformulated/re-thought.
Author Response
- As the Authors reported in the Manuscript, production of ATP in microorganisms is highly affected by the culture media. However, the examples reported in the Manuscript concern only in vitro cultured bacteria/fungi. Since in chapter 4, the Authors reported the role of ATP as inter-kingdom signalling molecule, the Authors should report also some evidences about production of ATP by wild-type microorganisms in their natural environment.
Response 1: We absolutely agree with this major concern but unfortunately, there is no publication available that has specifically addressed this question in a living organism until now. Research on bacterial ATP is still in its infancy. Studies in vivo have the main limitation that ATP is chemically identical whether it is released by the microorganisms or by the host. This means that either the microorganisms have to be genetically modified (e.g. with a plasmid bearing an apyrase like the Grassi group did, ref. 77, 83, 85, lines 221-231) to investigate specifically bacterial ATP in vivo. Otherwise only total ATP can be assessed (e.g. in the gut like Inami did in his review, ref. 28). However, to assess total ATP is tricky because there are numerous interactions between the microorganisms and the host (e.g. stimulation of enterocytes to secrete ATP by bacteria ref. 42, lines 252-262 or the presence of ectonucleotidases, see below). Therefore, it is virtually impossible to determine the origin (bacteria, enterocytes, lamina propria cells) or the amount of bacterial ATP if total ATP from a natural environment is assessed. In chapter 4, we summarized all the available knowledge about physiological or pathological changes in the host that can be specifically attributed to bacterial ATP.
2. The Authors reported that proximity of bacteria and host cells is pivotal in determining the signal mediated by ATP. Since both microbial and eukaryotic cells are endowed with enzymes involved in ATP degradation, could the Authors make comments about the balance (concentration, time) between degradation and activity of ATP during inflammation?
Response 2: Eukaryotic cells ubiquitously express NTPDases (e.g. CD39), ecto-5’-nucleotidases (e.g. CD73), ectonucleotide pyrophosphatases and alkaline phosphatases on the surface. In peritoneal inflammation, the amount of extracellular ATP is in the low nanomolar range (Dosch et al., ref. 30) and there is a very informative review from Zimmermann et al. (“Cellular function and molecular structure of ectonucleotidases”), which shows that ATP in high micromolar concentrations is hydrolyzed within 20-40 min. ATP is therefore generally considered as a fast and short-term signaling molecule. This reference and a short comment have been added to the review (lines 245-251).
Prokaryotes have alkaline phosphatases and 5’-nucleotidases in the periplasmic space, which hydrolyze eATP in a linear concentration-dependent manner (lines 134-143). They also have ATP-hydrolyzing enzymes on the surface and this studies have been added to section 4 (lines 278-284). Unfortunately, these studies only describe the effect of this surface enzymes during inflammation but not the specific kinetics.
3. Purinergic signal, DAMP and inflammation are usually considered all together. However, ATP is physiologically produced also by non-pathogenic bacteria such as in the gut microbiome. The Authors should refer also to ATP produced by symbiotic bacteria when they consider the ATP as an inter-kingdom signalling molecule.
Response 3: We fully agree that ATP produced by the microbiome has to be mentioned in this review despite the scarce literature. However, we did mention the specific role of bacterial ATP on host IgA production in the gut and subsequent decreased efficiency of oral vaccination (lines 220-230). We also mentioned that bacterial ATP limits IgA production by the host, which has a favorable metabolic effect on blood sugar levels and body weight (lines 220-230). Furthermore, we mentioned the findings from Atarashi et al., which show the importance of commensal bacteria-derived ATP on physiologic and pathologic TH17 differentiation (lines 232-242). Since our aim was to summarize everything that is known in vitro as well as in vivo specifically attributable to bacterial ATP, we did not want to include findings about general extracellular ATP in the gut.
4. The Authors should better define what they intend as “luminal ATP” referred to Caco-2 cells experiment (lines 241-242)
Response 4:In this in vitro experiment luminal ATP was wrongly used. It is about apical ATP on a polarized monolayer of Caco-2 cells. The sentence was corrected accordingly (lines 252-254).
5.Lines 43, 66, 218, 219, 253,260: replace “EATP” with “Extracellular ATP”, for clarity
“EATP” has been replaced by “Extracellular ATP” if it is at the beginning of the sentence
6. In the legend of Figure 1, I suggest inserting the correspondent citation for references for each described point (A-F)
The references for the described points have been inserted into the figure.
7.Sentence lines 181-182 need to be reformulated/re-thought
The sentence has been complemented to point out the relatedness to the paragraph above.
Reviewer 2 Report
This review provides a detailed overview of the release of ATP from bacteria, including its mechanisms of release and impact on host responses. This review is highly novel with, to the best of my knowledge, none to date on this topic. The quality of the writing and the one figure is excellent. I have only minor comments.
- On lines 105-110 the authors discussed differences between studies of bacterial ATP release. Can the authors confirm that these differences were not due to the presence of serum in the media, which may have also accounted for these differences and if so modify the text accordingly? Also define "RPMI" more fully (line 86ff), such as "RPMI 1640" or as appropriate, although I appreciate some companies are starting to use "RPMI" only.
- The authors may wish to note briefly in Section 3.1 (or elsewhere if more appropriate) that bacteria or their products can induce ATP release from eukaroytic cells to stimulate P2 receptors as demonstrated by HA Praetorius et al and others.
- Some nomenclature can be improved. (A) "adenosine-5'-triphosphate" (lines 12 and 26) should be "adenosine 5'-triphosphate" (omit first "-"). Likewise the same for other nucleotides thereafter on lines 41 ff (ADP, UTP etc) for example. (B) According to international IHPHAR/BPS guidelines, numbers for P2X receptors should not be subscripted; subscript numbers only for the GPCRs P1 and P2Y receptors. (C) Italicize "bo" in cytochrome bo" (lines 122ff). (D) Correct "toll" to "Toll" (line 272). (E) Delete "^" and superscript numbers accordingly (lines 94, 241, 243 and possibly others).
- Titles of articles in references should be correctly capitalized. Many are in sentence case.
Author Response
- On lines 105-110 the authors discussed differences between studies of bacterial ATP release. Can the authors confirm that these differences were not due to the presence of serum in the media, which may have also accounted for these differences and if so modify the text accordingly? Also define "RPMI" more fully (line 86ff), such as "RPMI 1640" or as appropriate, although I appreciate some companies are starting to use "RPMI" only.
Response 1: As suggested, RPMI medium has been changed in RPMI 1640 medium. According to the authors of the respective studies, these are all in vitro studies and neither RPMI 1640 medium nor LB medium should be enriched or be contaminated with serum. A sentence was added to clarify this in the review (lines 105-108).
- The authors may wish to note briefly in Section 3.1 (or elsewhere if more appropriate) that bacteria or their products can induce ATP release from eukaroytic cells to stimulate P2 receptors as demonstrated by HA Praetorius et al and others.
Response 2: In section 4 “EATP as an inter-kingdom signaling molecule” there is a small paragraph about E. coli-mediated ATP release from rat jejunum and colon adenocarcinoma-derived Caco-2 cells. We completed this paragraph with some findings from Praetorius et al. (line 256-261).
- Some nomenclature can be improved. (A) "adenosine-5'-triphosphate" (lines 12 and 26) should be "adenosine 5'-triphosphate" (omit first "-"). Likewise the same for other nucleotides thereafter on lines 41 ff (ADP, UTP etc) for example. (B) According to international IHPHAR/BPS guidelines, numbers for P2X receptors should not be subscripted; subscript numbers only for the GPCRs P1 and P2Y receptors. (C) Italicize "bo" in cytochrome bo" (lines 122ff). (D) Correct "toll" to "Toll" (line 272). (E) Delete "^" and superscript numbers accordingly (lines 94, 241, 243 and possibly others).
Response 3: The dashes have been removed. (B) The numbers for the P2X receptors have been corrected according to the international IHPHAR/BPS guidelines. (C) bo is italicized. (D) Toll is corrected. (E) Numbers are superscript now.
4. Titles of articles in references should be correctly capitalized. Many are in sentence case.
Response 4: All references are inserted once again with Endnote and the MDPI citation style and manually controlled.
Round 2
Reviewer 1 Report
none